# Determination of 2-Acetyl-1-pyrroline via a Color-Change Reaction Using Chromium Hexacarbonyl

**DOI:** 10.3390/molecules27123957

**Published:** 2022-06-20

**Authors:** Chonlada Bennett, Woraprapa Sriyotai, Sirakorn Wiratchan, Natthawat Semakul, Sugunya Mahatheeranont

**Affiliations:** 1Department of Chemistry, Faculty of Science, Chiang Mai University, Chiang Mai 50200, Thailand; chonlada8@gmail.com (C.B.); woraprapa_s@cmu.ac.th (W.S.); sirakorn_w@cmu.ac.th (S.W.); 2Research Center on Chemistry for Development of Health Promoting Products from Northern Resources, Chiang Mai University, Chiang Mai 50200, Thailand; 3Center of Excellence for Innovation in Chemistry, Faculty of Science, Chiang Mai University, Chiang Mai 50200, Thailand

**Keywords:** 2-Acetyl-1-pyrroline, chromium hexacarbonyl, color-change reaction, colorimetry, fragrant rice, gas chromatography–mass spectrophotometry

## Abstract

At present, there is no colorimetric method for the quantitation of the aroma compound 2-acetyl-1-pyrroline (2AP). A novel colorimetric method was developed for the determination of 2AP content using chromium hexacarbonyl (Cr(CO)_6_) as a reagent. The reaction of synthetic 2AP with chromium hexacarbonyl reagent solution in the presence of light produced a green product with an absorption maximum (λ_max_) at 623 nm. GC–MS was used to confirm the color-change reaction, which showed the loss of 2AP after the addition of Cr(CO)_6_. This novel method enables facile and cost-effective determination of 2AP in fragrant rice. A comparative analysis of fragrant and nonfragrant rice grain extracts showed that no color-change reaction occurred with the nonfragrant rice sample. A limit of detection (LOD) of 2.00 mg L^−1^ was determined by method validation with an effective linear concentration ranging from 5.00 to 60.00 mg L^−1^ of 2AP. The results obtained using the developed colorimetric method were consistent with those obtained by automated static headspace gas chromatography with nitrogen-phosphorus detection (SHS-GC–NPD).

## 1. Introduction

2-Acetyl-1-pyrroline (2AP) is a prominent aroma compound found in many foods that is described as smelling ‘popcorn-like’. 2AP is a five-membered N-heterocyclic ring compound, which makes the compound highly volatile and unstable with a low odor threshold. 2AP has a boiling point of 183 °C and a molecular weight of 111.14 [1]. 2AP contributes to the flavor of a diverse array of foods, such as cooked mushrooms, nuts, sweet corn, and lobster [2]. 2AP has many functions, for example, providing aroma in essential oils used as therapeutic fragrances and acting as a flavoring compound or food additive [3]. 2AP is biosynthetically metabolized in several plant species, including soybean (Glycine max), sorghum (*Sorghum bicolor* (L.) Moench), fragrant coconut (*Cocos nucifera* L.), winter melon (*Benincasa hispida*) and yellow dwarf coconut (*Cocos nucifera* L.) [4,5,6,7,8]. Fragrant or aromatic rice contains over 200 volatiles with 2AP as the major component for the pleasant aroma. This quality is highly regarded for its superior grain quality [1]. 2AP can also be found in wheat, brown and rye bread, where the 2AP content of the crust is reported to be 30-fold that of the crumbs [9].

The development of analytical methods for the determination of 2AP in food samples in recent years has been challenging because of the volatility and low concentration (in some samples) of 2AP, as well as the high cost of high-purity chemical standards [10]. The determination of 2AP content is usually carried out via gas chromatography (GC) employing various detection systems, such as nitrogen-phosphorus detectors (NPD), flame ionization (FID), and mass spectrophotometry (MS) [11,12]. Additionally, static headspace (SHS) or solid-phase microextraction (SPME) coupled with GC–MS methods has proven to be a powerful tool for the detection of 2AP at very low concentrations (ppb) [13]. However, GC analysis is extremely time consuming, expensive and requires trained personnel.

There have been no reports on the qualitative or quantitative analysis of 2AP via colorimetry to date. Nadaf et al. reported that spraying a 2,4-dinitrophenyl hydrazine reagent onto a TLC plate loaded with 2AP extracted from pandan leaves produced an orange-red compound [14]. The color-change reaction was identified as that between the reactive ketone group in 2AP and 2,4-dinitrophenyl hydrazine to produce the colored compound hydrazone derivative. Other detection methods based on colorimetric sensor assays have been developed and validated for the analysis of volatile compounds in rice samples. Data processing or principal component analysis is often used to classify volatile compounds to predict the desired outcome in an assay. These outcomes include the differentiation of different rice cultivars or rice storage times [15,16]. Chemically responsive dyes have been used to visualize volatile compounds by recording the change in the dye color before and after exposure to the volatile compounds. The aforementioned colorimetric sensors offer many advantages for qualitative analysis and quantitative discrimination, but their analytical power depends on information obtained using other methods. The qualitative and quantitative analytical capacities of these methods are therefore limited. Consequently, a simple colorimetric method was developed for the quick and easy quantitation of 2AP content via UV–Vis spectrophotometry.

Buttery and de Kimpe reported chemical synthesis methods for preparing 2AP for use as a standard compound [17,18]. In this study, 2AP was chemically synthesized through hydrogenation and oxidation reactions using commercially available 2-acetylpyrrole as a precursor. A colorimetric method was then developed by investigating the color change reactions for different types of organic compounds and organometallic complexes with 2AP. The developed colorimetric technique can be applied to analyze 2AP in fragrant rice samples without the use of advanced analytical instruments, such as GC or GC–MS. The proposed method is applicable and can easily be adapted to a diverse number of samples. The novel method exhibits comparable analytical performance to chromatographic-based methods, cost-effective and time-efficient analysis. Specialized training is not required to perform this colorimetric method, which can therefore be used by farmers and people in the private sector who are concerned with the aroma compound 2AP.

## 2. Results and Discussion

### 2.1. Synthetic 2AP and Its Reaction with Reagents

Synthetic 2AP dissolved in dichloromethane (DCM) was screened using several reagents, and the UV–Vis absorption spectra of the reagent and the colored products were recorded using a UV–Vis spectrophotometer. The λ_max_ values of the reaction product and the colors produced are summarized in Table 1. Distinct absorption bands for the product of the reaction with 2AP were observed for some of the aromatic amine and aromatic hydrazine reagents, such as aniline hydrochloride, aniline sulfate, phenylhydrazine hydrochloride and 2,4-dinitrophenylhydrazine. For example, λ_max_ was 232 and 272 nm for phenylhydrazine hydrochloride but 368 nm for phenylhydrazine condensed with synthetic 2AP. It has been suggested that 2AP interacts more effectively with aromatic amines in salt form, such as aniline hydrochloride or aniline sulfate, than aniline and *p*-nitroaniline. Aromatic amines are more soluble in salt or acidic environments in which the protonation of –C=O increases the positive charge of C, thereby accelerating the nucleophilic attack of aromatic amines on the ketone functional group. The λ_max_ values were 203–205 nm and 253 nm for aniline hydrochloride and aniline sulfate, respectively, and changed to 378 nm after these compounds reacted with 2AP. The nitro group at the ortho and para positions of 2,4-dinitrophenylhydrazine stabilizes the overall structure, facilitating the reaction [19]. Furthermore, an λ_max_ of 380 nm was observed for 2,4-dinitrophenylhydrazine reacted with 2AP, whereas an λ_max_ was not detected for the pure reagent solution. Overall, aniline hydrochloride, aniline sulfate, phenylhydrazine hydrochloride and 2,4-dinitrophenylhydrazine exhibited low specificity for 2AP in rice grain extracts.

An absorption band with an λ_max_ of 430 nm was observed for 2-aminobenzaldehyde, whereas a distinct absorbance with an λ_max_ of 623 nm was observed for Cr(CO)_6_ and 2AP. Cr(CO)_6_ exhibited the highest potential of the reagents in the organometallic group due to the formation of a bright green product after exposure to light, whereas no absorbance for Co_2_(CO)_8_ was detected either for the reagent solution or upon reaction with synthetic 2AP. In comparison to the results for Cr(CO)_6_, the product obtained using 2-aminobenzaldehyde had a light yellow color that was difficult to detect with the naked eye. The reaction of 2-aminobenzaldehyde with the fragrant rice grain extract in DCM exhibited a similar absorbance (λ_max_ = 423 nm) as the synthetic 2AP (λ_max_ = 427 nm), as expected, but the intensity was low. Note that 2-aminobenzaldehyde is an expensive and unstable reagent that must be stored below −20 °C [20]. By contrast, Cr(CO)_6_ is inexpensive, stable at room temperature, and dissolves effectively in various organic solvents. Therefore, Cr(CO)_6_ was the reagent of choice for the colorimetric method.

### 2.2. Confirmation of the Interaction between 2AP and Cr(CO)_6_ via GC–MS

A possible mechanism for the formation of the green product of the light-assisted reaction between Cr(CO)_6_ and 2AP is shown in Figure 1. The compound 2AP could be considered a chelating ligand through the N and O atoms [21,22]. When activated by light, the CO ligands dissociate to expose vacant coordination sites on the chromium, facilitating the association of 2AP to produce a green Cr(CO)_6-n/2_(2AP)_n_ complex. In this case, a substitution mechanism of CO by 2AP cannot be ruled out. Control experiments were carried out to elucidate complex formation. First, the green solution was not obtained in the absence of light, implying that light is crucial for complexation. Next, reacting Cr(CO)_6_ with 2-acetylpyrrole failed to produce a green solution upon exposure to light. This result was attributed to the inability of the lone pair of electrons in pyrrolic N to serve as a good donor atom because of its participation in the aromatic ring. However, light-assisted complexation of Cr(CO)_6_ and 2-acetylpyridine did produce a green product (λ_max_ = 606 nm), albeit with low molar absorptivity, indicating that the pyridinic N is a less effective ligand than 2AP [23].

Figure 2 shows the GC–MS chromatograms of synthetic 2AP in toluene (Figure 2A) and DCM (Figure 2B) before and after the addition of Cr(CO)_6_ (Figure 2C). These chromatograms show that 6-methyl,5-oxo-2,3,4,5-tetrahydropyridine (6M5OTP) was not present in the synthetic 2AP sample, where 2AP was formed by the oxidation of 1-(pyrrolidin-2-yl)ethan-1-ol with silver (I) carbonate on celite in toluene. In our experiment, 6M5OTP formed in relatively high quantities during acidic extraction to transfer 2AP from toluene to DCM. The presence of the 6M5OTP compound has been reported to correlate strongly with the production of 2AP and is simultaneously produced with 2AP in rice, contributing to the aroma of fragrant rice [24]. Olfactory analysis of 6M5OTP via gas chromatography has provided evidence of a similar scent to 2AP. The results of our studies showed that 6M5OTP did not interfere with light-assisted complexation with Cr(CO)_6_. GC–MS analysis revealed that the isomer 6M5OTP was consumed following light-assisted complexation, implying that 6M5OTP isomerizes to 2AP via an open chain structure. Then, thermodynamics drive subsequent complexation of 2AP with Cr(CO)_6_. Overall, the highly selective complexation of 2AP with Cr(CO)_6_ can be used to determine 2AP by a simple colorimetric method. The mass spectrum of 2AP, 6M5OTP and Cr(CO)_6_ are shown in Figure 2D–F, respectively.

### 2.3. Specificity of the Colorimetric Method for the Determination of 2AP

Fragrant rice samples extracted with 100.00, 150.00 and 200.00 g of rice grains showed similar absorbances at 623 nm. These results were consistent with those of the aforementioned experiment on synthetic 2AP at various concentrations. The GC–MS profiles of the fragrant rice extract before (Figure 3A) and after (Figure 3B) the addition of Cr(CO)_6_ confirmed that the reaction of Cr(CO)_6_ is specific to both 2AP and 6M5OTP. The reaction of Cr(CO)_6_ with fragrant and nonfragrant rice grain extracts was investigated. The absorption maximum at 623 nm was observed in the extracts of fragrant rice grains but not in the nonfragrant-cultivar extract. The specificity of the reaction was also determined by studying the reaction of Cr(CO)_6_ with other nitrogenous compounds with chemical structures similar to that of 2AP that are found in naturally occurring samples. These compounds include pyrrolidine, 2-acetylpyrrole, 2-acetylpyridine, 2-acetylthiosol and proline. The absorbance band of the nitrogen compounds, especially proline, did not occur in the λ_max_ range of 620–670 nm that was expected from the reaction of 2AP and Cr(CO)_6_. Pyrrolidine, 2-acetylpyrrole and 2-acetylthiosol exhibited no absorbance between 400 and 700 nm. An λ_max_ of 606 nm and very low molar absorptivity was observed only for 2-acetylpyridine. 2-Acetylpyridine has been detected in a scented Chinese fragrant rice, Xiangjing-8618 in only one study [25]. To the best of our knowledge, 2-acetylpyridine does not naturally occur in the seeds of fragrant or nonfragrant varieties of Thai rice. Two-dimensional gas chromatography (GC×GC) was used to confirm the presence of 2AP in fragrant and nonfragrant rice samples (Figure 4). GC×GC techniques can be a valuable tool to study the volatile fraction where every part of the sample is subjected to two individual separation dimensions. The qualitative analysis of volatile components in seed extracts of more than twenty rice varieties using GC×GC–MS did not identify 2-acetylpyridine in the samples, while the presence of 2AP in fragrant rice KDML105 and PSL80 cultivars were confirmed (Figure 4). One limitation of this developed colorimetric method is that some other aromatic plants or foods contain compounds with the same active site as 2AP that can interact with the reagent and cause an inaccurate result. Therefore, a good understanding of the chemistry of the sample of interest is considered a prerequisite for accurate detection and evaluation of aroma quality.

### 2.4. Analytical Performance and Validation

The effect of the following factors on the rate of the reaction between synthetic 2AP and the Cr(CO)_6_ reagent to produce a colored product was investigated: the light exposure time (5, 10, 15, 20, 25 and 30 min) and the distance from light source (10, 15 and 20 cm). The UV–Vis spectrum of the green product solution exhibited an λ_max_ of 623 nm, which did not change significantly with the light exposure time. However, the absorbance of the colored product increased with exposure time, where the green color appeared much darker at 30 min than at 5 min. The UV–Vis spectra at different 2AP concentrations are shown in Figure 5. The reaction between Cr(CO)_6_ and synthetic 2AP at concentrations below 15.00 mg L^−^^1^, the product solution appeared lighter in color than at concentrations above 15.00 mg L^−^^1^. The optimal reaction time was determined at 5 min where the green product decomposed into a green precipitate for reaction times of more than 15 min. Additionally, the distance of 10 and 15 cm of the product solution from the light source showed that precipitate was formed. The optimal distance was determined at 20 cm. 

The developed colorimetric method for 2AP quantitative analysis was validated with respect to the linearity range, sensitivity (limit of detection) and precision. The absorbance of the green product at 623 nm was linear in the synthetic 2AP concentration over the concentration range of 5.00–60.00 mg L^−^^1^ with a correlation coefficient (r^2^) of 0.9934. The sensitivity is reflected by the limit of detection (LOD), which is defined as the concentration at which the signal-to-noise ratio (S/N) is more than 3. The LOD of 2AP was found to be 2.00 mg L^−^^1^. The precision of the proposed method is expressed in terms of the repeatability (intraday) and reproducibility (interday) as a percentage of the relative standard deviation. A solution of synthetic 2AP at 40.00 mg L^−^^1^ in DCM was prepared and reacted with 1000 mg L^−^^1^ of Cr(CO)_6_ ten times on the same day under optimal conditions. The recorded absorbances at 623 nm were averaged, and the %RSD was calculated to be 4.31. The reproducibility was determined on five different days with five replicates on each day and yielded a coefficient of variation of 1.57 %RSD. The validation data showed lower sensitivity and precision for the colorimetric method than for automated headspace gas chromatography [11]; however, the data obtained using the colorimetric method are acceptable. Overall, although the multi-step sample preparation for the colorimetric method is time-consuming, the lack of sophisticated instrumentation results in a cost-effective analysis technique. 

Application of the colorimetric technique to the quantitation of 2AP in fragrant rice samples requires a properly designed experiment to ensure there is a sufficient quantity of extracted 2AP in DCM to react with Cr(CO)_6_ to yield a green product. An experiment was conducted using KDML105, the most popular Thai fragrant rice variety with a high content of 2AP. The optimal weight of the rice sample for extraction was found at 100 g.

### 2.5. Comparison of the 2AP Content Determined by SHS-GC–NPD and the Colorimetric Method

Extracts of fragrant rice grains and nonfragrant rice grains were quantitatively determined using the developed colorimetric method and SHS-GC–NPD (Table 2). The addition of Cr(CO)_6_ to extracts of KDML105 and PSL80 fragrant rice grains resulted in a color-change reaction, where a green solution was developed after exposing the solution to UV light. However, a color change was not observed for the nonfragrant rice grains PSL2 and RD79 upon addition of Cr(CO)_6_. These results were confirmed by SHS-GC–NPD, where the 2AP content of both cultivars could not be detected. SHS-GC–NPD indicated 3.68 ± 0.07 µg g^−^^1^ and the colorimetric method showed 3.83 ± 0.09 µg g^−^^1^ for KDML105 while those for PSL80 were 2.11 ± 0.14 µg g^−^^1^ and 1.95 ± 0.10 µg g^−^^1^ for colorimetric and SHS-GC–NPD, respectively. The PSL80 cultivar was reported as having been developed from KDML105 to contain less amylose but comparable aroma contents [26]. These results were obtained from a study published by the Rice Department of Thailand, in which the similarities between cultivars PSL80 and KDML105 were identified.

## 3. Materials and Methods

### 3.1. Chemicals and Reagents

Analytical grade chemicals, including 2-acetylpyrrole, 2-acetylpyridine, aniline, aniline hydrochloride, aniline sulfate, p-nitroaniline, p-nitroaniline hydrochloride, 2-aminobenzoic acid, 2-aminobenzonitrile, 2-aminobenzaldehyde, phenylhydrazine, phenylhydrazine hydrochloride, 2,4-dinitrophenylhydrazine, dicobalt octacarbonyl (Co_2_(CO)_8_), chromium hexacarbonyl (Cr(CO)_6_), 5% rhodium on an activated alumina (Rh/Al_2_O_3_) catalyst (99% purity) and internal standard 2,6-dimethylpyridine (2,6-DMP) were purchased from Sigma Aldrich, St. Louis, MO, USA. Silver nitrate (AgNO_3_) was purchased from POCH, Warsaw, Poland. Methanol (HPLC grade), dichloromethane (AR grade), hydrochloric acid (AR grade), toluene (AR grade) and sodium sulfate anhydrous (AR grade) were purchased from RCI Labscan, Bangkok, Thailand. Deionized water was obtained from a Nanopure Ultrapure water purification system (Thermo Fisher Scientific Inc., Waltham, MA, USA).

### 3.2. Synthesis of 2-Acetyl-1-pyrroline 

2-Acety-1-pyrroline was synthesized by the hydrogenation of 2-acetylpyrrole to yield 1-(pyrrolidin-2-yl)ethan-1-ol, which was then oxidized by silver (I) carbonate. The experimental method involved considerable modifications of methods reported in the literature [17,18] and is detailed below.

#### 3.2.1. Silver (I) Carbonate on Celite

Silver (I) carbonate on Celite was prepared using a method reported by Fétizon [27]. Celite (30.00 g) was added to a solution of AgNO_3_ (30.00 g of AgNO_3_ in 200 mL of distilled water) and vigorously stirred for 10 min. Then, a solution of Na_2_CO_3_ (30.00 g of Na_2_CO_3_·10H_2_O in 300 mL of distilled water) was slowly added to the suspension to yield a yellow-green precipitate of Ag_2_CO_3_ on Celite. The product was recovered by filtration and washed several times with distilled water until the pH reached ca. 7. The product was dried in an oven at 110 °C for 24 h.

#### 3.2.2. 1-(Pyrrolidin-2-yl)ethan-1-ol

A 250-mL round-bottomed flask equipped with a magnetic stir bar was filled with 2-acetylpyrrole (0.700 g) and methanol (50.00 mL). Then, Rh/Al_2_O_3_ (1.00 g) was added to the solution, followed by purging with H_2_ using a balloon. The reaction progress was monitored by TLC and GC–MS to ensure complete consumption of 2-acetylpyrrole. After 48 h, the solution was filtered through Celite and evaporated to dryness to afford 1-(pyrrolidin-2-yl)ethan-1-ol in quantitative yield. ^1^H-NMR and GC–MS analyses showed that the obtained compound was sufficiently pure to be used without further purification.

#### 3.2.3. 2-Acetyl-1-pyrroline (2AP)

A 250-mL round-bottomed flask equipped with a magnetic stir bar was filled with 1-(pyrrolidin-2-yl)ethan-1-ol (0.625 g), Ag_2_CO_3_ on Celite (5.00 g) and toluene (150.00 mL) under an N_2_ atmosphere. The reaction was heated to reflux for 1 h to obtain 2-acetyl-1-pyrroline. The suspension was filtered through Celite and transferred to a 250-mL volumetric flask to obtain a stock solution of 2AP in toluene. The 2AP in toluene was extracted twice with 100.00 mL of 0.1 M HCl. Then, the aqueous solution was basified with 1 M NaOH and extracted with DCM. A portion of the organic layers was transferred to a 1 L volumetric flask to obtain a stock solution of 250.00 mg L^−^^1^ 2AP in DCM.

### 3.3. Plant Materials and Extraction

Fragrant and nonfragrant rice grains were used to validate the color-change reaction between the naturally occurring 2AP in plant samples and Cr(CO)_6_. The fragrant rice grains used were *Oryza sativa* L. cv. Khao Dawk Mali 105 (KDML105) and Phitsanulok 80 (PSL80), and the nonfragrant rice grains used were *Oryza sativa* L. cv. Phitsanulok 2 (PSL2) and Rice Department 79 (RD79). Samples were extracted using the acid-base method. Briefly, 100.00 g of rice grains ground in a blender and added to 200.00 mL 0.1 M HCl; the resulting mixture was shaken for 15 min and centrifuged at 6000 rpm for 15 min. The supernatant was basified with approximately 2.00 mL of 1 M NaOH and added to 200.00 mL of DCM. The solution was shaken and left to stand until it separated into layers. The DCM portion was collected and added to anhydrous Na_2_SO_4_ to remove water molecules. The solution was dried using a rotary evaporator to obtain 2.00 mL of 2AP extract.

### 3.4. GC–MS Analysis

A Model 6850 gas chromatograph equipped with a Model 5973 mass spectrometer with dimensions of 30 m × 0.25 mm and a 0.25-μm nonpolar capillary column HP-5MS (5%-phenyl)-methylpolysiloxane (Agilent Technologies, Santa Clara, CA, USA) was used to analyze the synthetic 2AP, including its intermediate, byproducts and reaction product with Cr(CO)_6_. The GC carrier gas used was purified helium (99.99%) at a constant flow rate of 1 mL min^−^^1^. The oven temperature was initially held at 45 °C and then increased at a rate of 3 °C min^−^^1^ to a final temperature of 150 °C. To analyze the rice sample extracts, the initial oven temperature was set at 45 °C, held for 5 min and increased to 200 °C at a rate of 4 °C min^−^^1^. The injector temperature was 230 °C. The sample was injected with a split ratio of 10:1. The mass spectrometer was operated in electron impact (EI) mode with an electron energy of 70 eV. The ion source and quadrupole temperatures were set at 230 °C and 150 °C, respectively. The mass range (*m*/*z*) was 29–550 with a scan rate of 0.68 s. The GC–MS transfer line was set to 280 °C. 2AP and Cr(CO)_6_ were identified by matching their mass spectra with synthetic and standard compound reference spectra in the Wiley 7n Mass Spectral Library (Revision C.00.00). The NIST 17 Mass Spectral Library (Revision D.01.00/1.6d.) was also utilized.

### 3.5. Comprehensive Two-Dimensional Gas Chromatography–Mass Spectrometry (GC×GC–MS) Analysis

GC×GC–MS was carried out on an Agilent Technologies 6890 model GC system retrofitted with a longitudinally modulated cryogenic system (LMCS). The instrument was equipped with a headspace auto-sampler and a flame ionization detector. The column set-up consisted of primary capillary column dimensions of 30 m × 0.53 mm i.d. × 1.5 µm film thickness HP-5 phase (5% phenyl 95% methylpolysiloxane) coupled with a second capillary column dimensions of 1.0 m × 0.32 mm i.d.× 0.5 µm film thickness Innowax phase (Polyethylene glycol). The thermostatically controlled cryogenic trap was maintained at about −20 °C for the duration of each analysis. The modulation period was 6 s in all analyses. Temperature-programmed conditions were performed for all analyses, using an initial oven temperature of 45 °C (constant for 4 min), then increased at a rate of 3 °C per minute to a final oven temperature of 230 °C. The GC injector was operated in splitless mode, and the temperature was set at 230 °C. Purified helium gas was used as the carrier gas. The mass spectrometer was operated in electron impact (EI) mode with the same parameters as the above.

### 3.6. Chemical Reaction of 2AP with Reagents to form Colored Products

To identify reagents capable of forming colored products with 2AP, three groups of reagents were investigated that were classified according to their structure and reaction mechanisms. These groups included aromatic amines (aniline, aniline hydrochloride and aniline sulfate, p-nitroaniline, p-nitroaniline hydrochloride, 2-aminobenzoic acid, 2-aminobenzonitrile, and 2-aminobenzaldehyde); phenylhydrazines (phenylhydrazine, phenylhydrazine hydrochloride and 2,4-dinitrophenylhydrazine); and organometallic complexes (Co_2_(CO)_8_) and Cr(CO)_6_). Reagent solutions were prepared at a concentration of 1000 mg L^−^^1^ in distilled water or DCM. Then, 1.00 mL of each reagent was added to 100.00 mg L^−^^1^ of synthetic 2AP in DCM. The color-change reaction was allowed to occur under a solar light source (Hagen Inc., Baie-D’Urfe, QC, Canada) for 5 min, and the solution of the colored product was scanned using a spectrophotometer with a wavelength range of 400–700 nm. A 125-W Solar-Glo lightbulb was used as the solar light source and emitted UVA, UVB, and visible light.

### 3.7. Colorimetric Method

A series of 2AP standard solutions with concentrations of 5.00–60.00 mg L^−^^1^ in DCM was prepared by diluting from the stock solution of 250 mg L^−^^1^ 2AP in DCM described in 3.2.3. Each of the synthetic 2AP solutions of different concentrations (1.00 mL) was added to 1.00 mL of 1000 mg L^−^^1^ Cr(CO)_6_ reagent solution. The resulting mixture was shaken and exposed to the solar light source (125-W Solar-Glo lightbulb) for 5 min. Then, the absorbances at 623 nm were recorded. The colorimetric method was qualitatively validated in terms of specificity and sensitivity. Reagent specificity to the 2AP compound was assessed for fragrant and nonfragrant rice grain extracts, and other nitrogen-containing compounds, including 2-acetylpyrrole and 2-acetylpyridine, in terms of the presence of a colored product. The reagent sensitivity was assessed by the limit of detection (LOD) of the 2AP standard. GC–MS was used to monitor the formation of 2AP and its isomer 6-methyl, 5-oxo-2,3,4,5-tetrahydropyridine (6M5OTP), which can form during the acidic extraction of rice grains.

### 3.8. SHS-GC–NPD

The determination of 2AP in grains was carried out using the method of Boontakham [28] with some modifications. First, rice grains were ground into a powder that was sieved through an aperture with a 45-mesh size. Then, the rice powder (1.00 g) was weighed into a 20-mL headspace vial, to which 1.00 µL of 500.00 mg L^−^^1^ 2,6-DMP was added as an internal standard. The headspace vial was then sealed immediately with PTFE/silicone septum and aluminum caps prior to analysis by SHS-GC–NPD. A static headspace autosampler (TurboMatrix 40, PerkinElmer, Inc., Waltham, MA, USA) connected to a PerkinElmer Clarus 690 Series GC system coupled to an NPD detector was used. Separation was carried out using an Elite-5MS column (with a 30 m × 0.25 mm i.d. and 0.25-µm film thickness; PerkinElmer, Inc., Waltham, MA, USA) with splitless injection at 250 °C. The column temperature was programmed to increase from 45 °C to 125 °C at a gradient of 7 °C min^−^^1^. The headspace operating conditions for the rice grains were a 125 °C oven temperature and a 20 min vial equilibration time.

### 3.9. Statistical Analysis

Samples were extracted in triplicate (*n* = 3). The determination of the 2AP content via SHS-GC–NPD, GC–MS and colorimetric methods was carried out in triplicate, and the results were expressed as a mean ± standard deviation (SD). Statistical analysis was performed using SPSS software version 17.0 (SPSS Inc., Chicago, IL, USA) for the one-way analysis of variance (ANOVA). The significance level at 95% (*p* < 0.05) was determined by the post hoc Tukey’s test.

## 4. Conclusions

In this study, we developed a colorimetric method for the quantitation of the aroma compound 2AP. Synthetic 2AP and extracted 2AP from plant samples were added with Cr(CO)_6_ solution, and a bright green color change reaction occurred. A distinct absorbance with an λ_max_ of 623 nm was observed for the green color product solution. The method was validated with fragrant and nonfragrant rice extracts, as well as other nitrogen-containing compounds to confirm the reaction specificity of 2AP. A color-change was not observed for the nonfragrant rice grains PSL2 and RD79 upon addition of Cr(CO)_6_. The quantitative results were confirmed by SHS-GC–NPD, where the good agreement was obtained for the concentrations of 2AP in the fragrant and nonfragrant rice samples. The colorimetric method proposed in the manuscript is simple and easy to perform by untrained personnel. The results obtained will be of interest for researchers in the related field who can develop our method further, including other food samples and on-field applications.

## Figures and Tables

**Figure 1 molecules-27-03957-f001:**
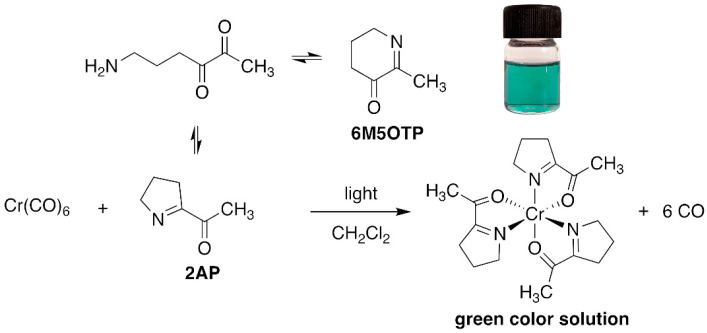
Possible mechanism for the light-assisted complexation of Cr(CO)_6_ and 2AP.

**Figure 2 molecules-27-03957-f002:**
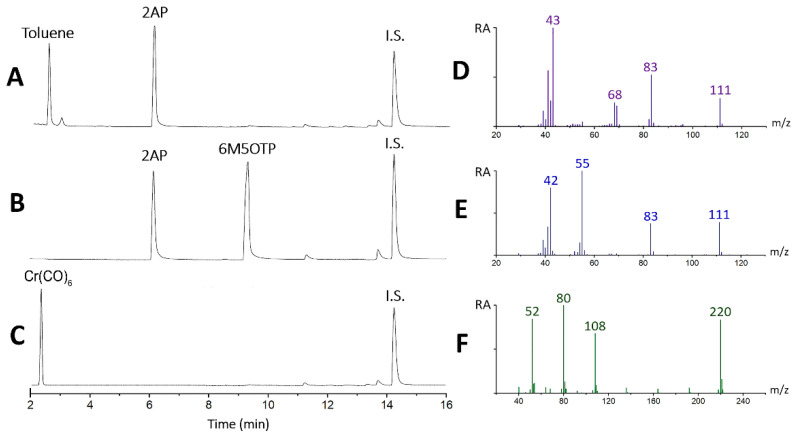
GC–MS chromatogram of synthetic 2AP in toluene (**A**), DCM (**B**) and after the addition of Cr(CO)_6_ (**C**). Mass spectrum of 2AP (**D**), 6M5OTP (**E**) and Cr(CO)_6_ (**F**).

**Figure 3 molecules-27-03957-f003:**
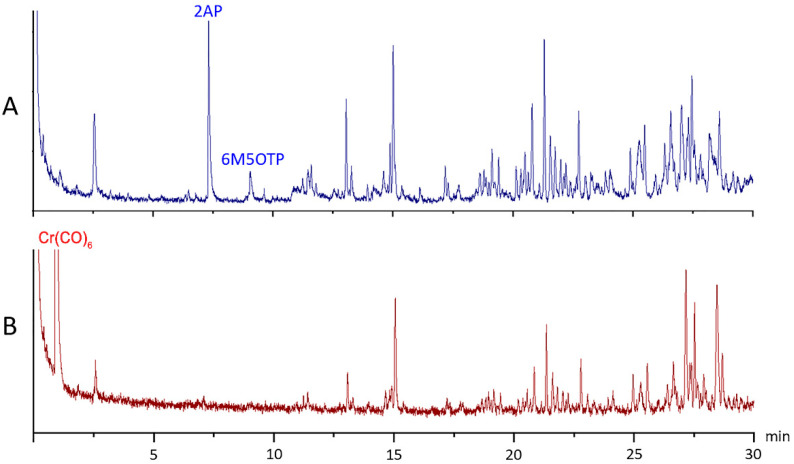
GC–MS chromatogram of a fragrant rice extract before (**A**) and after (**B**) the addition of Cr(CO)_6_.

**Figure 4 molecules-27-03957-f004:**
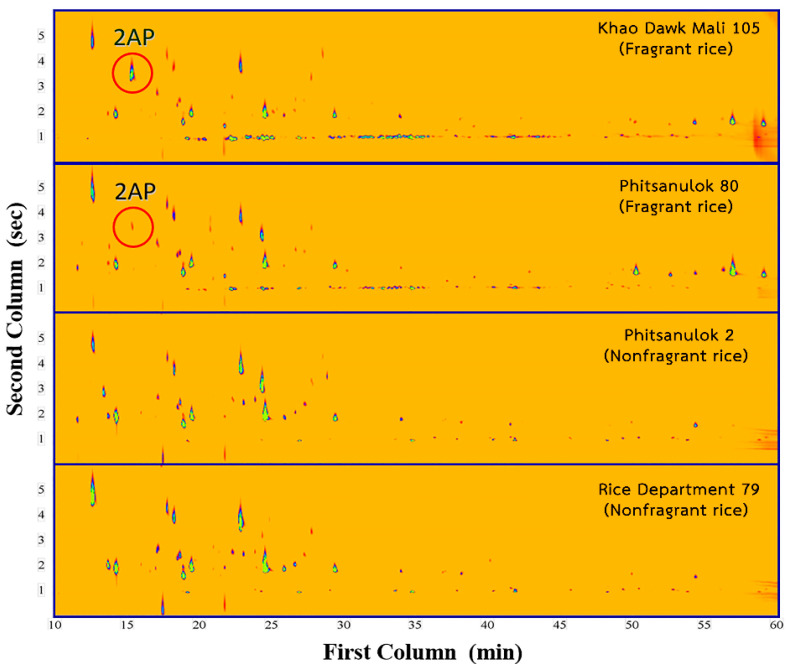
GC×GC–MS contour plots of extracts from fragrant and nonfragrant rice samples.

**Figure 5 molecules-27-03957-f005:**
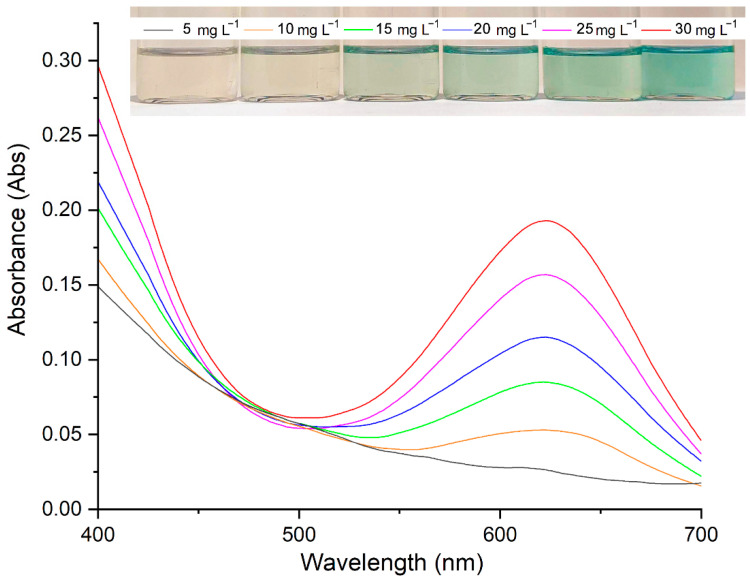
UV–Vis spectra of the colored product obtained from the reaction of Cr(CO)_6_ with 2AP at different concentrations.

**Table 1 molecules-27-03957-t001:** Reagents reacted with 2AP for the study of colored products.

Reagent	Structure	λ_max_ (nm) ofthe Colored Product	Product Color
Aniline	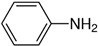	244	-
Aniline hydrochloride	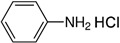	254	-
Aniline sulphate	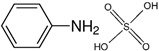	253	-
*p*-Nitroaniline	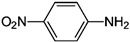	226	-
*p*-Nitroaniline hydrochloride	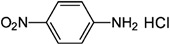	255	-
Phenyl hydrazine	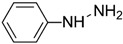	244	-
Phenyl hydrazine hydrochloride	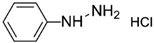	230	-
2,4-Dinitrophenyl hydrazine	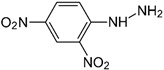	380	-
2-Aminobenzoic acid	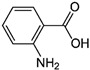	342	-
2-Aminobenzonitrile	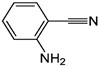	314	-
2-Aminobenzaldehyde	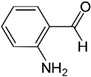	430	Light yellow
Dicobalt octacarbonyl (Co_2_(CO)_8_)	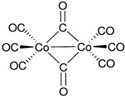	-	-
Chromium hexacarbonyl (Cr(CO)_6_)	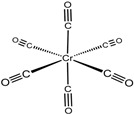	623	Green

**Table 2 molecules-27-03957-t002:** Quantitative analysis of 2AP in fragrant and nonfragrant rice samples.

Sample	Concentration of 2AP, µg g^−1^ (Mean ± SD)	
	Colorimetry	SHS-GC–NPD
Khao dawk mali 105 (KDML105)	3.83 ± 0.09	3.68 ± 0.07
Phitsanulok 80 (PSL80)	2.11 ± 0.14	1.95 ± 0.10
Phitsanulok 2 (PSL2)	ND	ND
Rice department 79 (RD79)	ND	ND

## Data Availability

Not applicable.

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
