# Peer review of "Determination of 2-Acetyl-1-pyrroline via a Color-Change Reaction Using Chromium Hexacarbonyl"

_molecules, 2022, doi:10.3390/molecules27123957_

Round 1
Reviewer 1 Report
Comments on manuscript titled “Determination via a color-change reaction using chromium heacetyl-1-pyterminatimon of 2-xacarbonyl”. Even though the sample preparation for 2AP determination in rice grain by colorimetric method is time consuming, the advantage is that no sophisticated instrumentation is required. However, methodology for colorimetric method (section 3.7) was not clearly described. Amount of sample, amount of reagent, type of solar light source, and absorbance wavelength should be revealed.
Overall, the manuscript was well written. Here are some specific suggestions:
Line 12, “there are no reports of a colorimetric…” delete “reports of a”
Line 13, change “discovered” to “developed”
Line 16, change “colour” to “color” “Both are correct. Pick one kind of spelling and stick with it throughout entire paper
Line 20, change “An LOD” to “A limit of detection (LOD)”
Line 35, remove “naturally”
Line 14, change “carried out” to “determined”
Line 47, remove “for operation”
Line 65, delete “have”
Line 68, revise the sentence “Then, a colorimetric…” to “A colorimetric method was then developed…”
Line 75, change “applied” to “used”
Line 79, change “dissolved in DCM” to “dissolved in dichloromethane (DCM)”
Line 94, change “drazine was expected to stabilize…” to “drazine stabilizes …”
Line 103, change Co2(CO)8 to Co2(CO)8
Line 115, delete “to be”
Line 190 and others, the word “min” should be changed to “minutes” make sure to change to minutes throughout the manuscript.
Line 193, change “with the exposure time” to “with exposure time”
Line 199, change “under less than 5 min” to “in less than 5 minutes”
Line 221, change “samples requires an experiment be properly designed” to “sample requires a properly designed experiment”
Line 239- 240, “The PSL80 cultivar…aroma contents”. Reference(s) is needed for this statement.
Line 250, change Co2(CO)8 to Co2(CO)8
Line 251, change “(Rh/Al2O3) to “(Rh/Al2O3)”
Line 253, change “(AgNO3)” to “(AgNO3)”
Line 294, How many mL of 0.1 M HCl?
Line 340, change Co2(CO)8 to Co2(CO)8
Line 381, change “illustrated” to “developed”
Author Response
Dear Sir/Madam,
Thank you very much for your comments on the manuscript entitled “Determination of 2-acetyl-1-pyrroline via a color-change reaction using chromium hexacarbonyl”, manuscript ID: Molecules 1762467 for publication in Molecules.
We carefully considered your comments and agreed with all of the points. Our point-by-point response to your comments and the changes we have made are summarized in the “Itemized list of changes Molecules 1762467_Reviewer 1” file attached.
We would like to take this opportunity to express our sincere thanks for your time identifying the areas of our manuscript that needed corrections or modification. Your kind support on the revised manuscript would be greatly appreciated.
Sincerely Yours,
Sugunya Mahatheeranont
Submitting author

Reviewer 2 Report
It is an interesting topic to found up an identification method between aromatic rice and non-aromatic rice via colorimetry to date in the manuscript. 2AP is suitable as characteristic compound of aromatic rice for producing a green complexation light-assisted by the method of color-change reaction between 2AP and Cr(CO)6, which is suitable of developing a colorimetric technique replacing GC or GC MS. The proposed method is applicable and can easily be adapted to a diverse number of samples. Therefore, it's reasonable for the colorimetric method that Cr(CO)6 is chosen as the reagent. The research on this method is reasonable, but in order to further explore the relevant reaction principle, it is hoped to supplement the possible corresponding analysis or detection. There are two specific points as follow,
1. The possible combination of Cr(CO)6 and 2AP should be analyzed from the theoretical calculation by using the method of metrology, so as to verify the feasibility of forming complexation ofCr(CO)6 and 2AP.
2. In view of the optimal reaction time was determined at 5 mins where the green product decomposed into a green precipitate for reaction times of more than 15 min. This precipitate also formed at high 2AP concentrations under less than 5 min of light exposure. Therefore, it is recommended to isolate the precipitate for identification.

Author Response
Dear Sir/Madam,
Thank you very much for your comments on the manuscript entitled “Determination of 2-acetyl-1-pyrroline via a color-change reaction using chromium hexacarbonyl”, manuscript ID: Molecules 1762467 for publication in Molecules.
We carefully considered your comments and agreed with all of the points. Our point-by-point response to your comments and the changes we have made are summarized in the “Itemized list of changes Molecules 1762467_Reviewer 2” file attached.
We would like to take this opportunity to express our sincere thanks for your time identifying the areas of our manuscript that needed corrections or modification. Your kind support on the revised manuscript would be greatly appreciated.
Sincerely Yours,
Sugunya Mahatheeranont
Submitting author

Reviewer 3 Report
This paper uses the color reaction between the carbonyl structure in 2AP structure and aromatic amines, phenylhydrazine, and organometallic compounds to establish a simple, time-saving, and efficient determination of 2AP content by colorimetry, which has certain practical significance. Some questions and suggestions for this article are as follows:
1. The colorimetric 2AP content determination method established in the article uses fragrant rice as the test object for method verification, but the introduction of plants and foods containing 2AP does not introduce the existence of 2AP in fragrant rice.
2. In order to verify the sensitivity of the colorimetric method, the article carried out a limit of detection (LOD) determination. As mentioned in the article, 2AP is present in a small amount in some samples and the final fragrant rice and non-fragrant rice determination results are related to the ability to measure very low concentrations. The results of static headspace gas chromatography-mass spectrometry (SHS-GC-MS) were compared and therefore 2AP colorimetry was suggested as a quantitative method to increase the limit of quantification for the determination.
3. Whether the illumination time means the reaction time? In the middle of the illumination time, rows 194-196 insert the exploration of different 2AP concentrations, but the results are not explained. At 5 mg/mL, the optimal wavelength of 623nm is no longer obvious. At the same time, it is shown below that when the 2AP concentration is high, a precipitate occurs in less than 5 min. Therefore, the colorimetric quantification of the concentration range of 2AP needs to be given in the text. The 2AP concentration used in the latter method validation was 40 mg/mL, but this level was not examined in the previous concentration discussions.
4. Figure 2D, 2E, and 2F gas mass spectrograms do not analyze the mass spectral data and what the mass spectra are used to illustrate.
5. The author's article indicates that the application of colorimetry to 2AP quantification needs to ensure that a sufficient amount of 2AP is extracted. The sample size of fragrant rice reaches 100 g. Is this the case for gas chromatography analysis?
6. In this paper, 2AP is obtained by synthesis to establish a standard curve. Whether the purchase of 2AP standards faces the problem of expensive high-purity standards mentioned in the introduction.
7. When screening colorimetric reagents with 2AP, the author indicated in line 334 that the reagents were divided into four groups according to their structure and reaction mechanism, but only aromatic amines, phenylhydrazine and organic compounds were introduced in lines 336-339.
8. There are inconsistencies in the number of decimal places reserved in the text, such as the linear range in the abstract and the expression of the linear range in lines 205 and 348, and similar problems in the text need to be further checked.
9. Indicate the full name of the first occurrence of the word, for example, DCM is not indicated, and LOD in the abstract
10. Unit and molecular formula problems, such as subscripts for units and molecular formulas in lines 103 and 307, NaSO4 in line 298, CrCO6 in Figure 2 and Figure 3
Author Response
Dear Sir/Madam,
Thank you very much for your comments on the manuscript entitled “Determination of 2-acetyl-1-pyrroline via a color-change reaction using chromium hexacarbonyl”, manuscript ID: Molecules 1762467 for publication in Molecules.
We carefully considered your comments and agreed with all of the points. Our point-by-point response to your comments and the changes we have made are summarized in the “Itemized list of changes Molecules 1762467_Reviewer 3” file attached.
We would like to take this opportunity to express our sincere thanks for your time identifying the areas of our manuscript that needed corrections or modification. Your kind support on the revised manuscript would be greatly appreciated.
Sincerely Yours,
Sugunya Mahatheeranont
Submitting author

Round 2
Reviewer 2 Report
The author has made some modifications to the suggestions, and has given a logical explanation to the contents that could not be modified. And the writing quality has been substantially optimized and improved. Therefore, it is recommended to give the acceptance for publishing.